# Modelling donor factors influencing pancreas transplant utilization and evolution of decision-making over time
Chahana Patel [1,2,3] ✉, Georgios Kourounis[1,2,3], Leonie van Leeuwen[4], Matthew Holzner[4], Vikram Wadhera[4], Mohammed Zeeshan Akhtar[4], Sander Florman[4], Angeles Maillo-Nieto[2], James Shaw[1,2,3], Steven White[1,2], Colin Wilson[1,2,3,5] & Samuel Tingle[1,2,3,5]

## Abstract

**Background** Pancreas transplantation remains the only definitive treatment for diabetes mellitus. However, the global number of pancreas transplants and utilisation of pancreas grafts is declining. We aimed to identify significant donor factors associated with pancreas non-use.

**Methods** Population-cohort study using United States (US) data from the Organ Procurement and Transplant Network (OPTN) registry (2010-2024). Multivariable regression models were constructed to assess associations between donor characteristics and pancreas utilisation. Restricted cubic splines were used to preserve non-linear relationships and interaction terms with donation date were performed, to capture evolving decision-making behaviours.

**Results** We identify 23 donor factors significantly associated with utilisation ($n = 14,612$ transplants from 133,986 donors). The most important continuous donor factors are age, BMI and peak creatinine; all showing significant non-linear relationships with utilisation (all $P < 0.001$). Donor type is the most important categorical variable, with donation after circulatory death (DCD) having 92% lower odds of utilisation (aOR=0.078, 95% CI = 0.070 to 0.087, $P = < 0.001$). Interaction analyses reveal increasing reluctance to use DCD donors or older donors over the study period (both interaction $P < 0.001$). Conversely, clinicians have become more comfortable transplanting pancreases from Hepatitis C positive donors and IV drug use (IVDU) donors over time (both interaction P < 0.001).

**Conclusions** This large population cohort study demonstrates significant shifts in utilisation decision-making over time. Growing reluctance to use DCD, despite evidence of favourable outcomes, highlights a valuable area to focus US pancreas utilisation efforts. Meanwhile, previously underused groups such as Hepatitis C positive and IVDU donors show growing acceptance, supporting expansion of these donor populations globally.

## Plain language summary

Pancreas transplantation can be a definitive treatment for diabetes, yet the number of pancreas transplants continues to decline, with many donated pancreases not being used. In this study, we analyse US data from 2010 to 2024 to determine which donor characteristics influence whether a pancreas is used. Using statistical models, we show certain donor factors, such as age and donor type, strongly impact whether the pancreas is used for transplant. We also find decision-making has evolved over time, with clinicians becoming increasingly accepting of certain donor groups, while becoming more reluctant to use others. These trends highlight important areas to focus efforts to expand the donor pool and improve access to pancreas transplantation, both in the US and globally.

For patients with type 1 diabetes mellitus, and selected patients with type 2 diabetes, pancreas transplantation is the only definitive treatment for maintaining euglycaemia and reducing long-term systemic complications[1]. Pancreas transplant numbers have seen a steady decline across the US and Europe, despite offering a significant improvement in quality of life[2]. There is rising global interest in improving pancreas utilisation and consequently pancreas transplant rates; however, there is a lack of high-quality research in this area.

[1]NIHR Blood and Transplant Research Unit, Newcastle University and Cambridge University, Newcastle upon Tyne, UK. [2]Institute of Transplantation, The Freeman Hospital, Newcastle upon Tyne, UK. [3]Translational and Clinical Research Institute, Newcastle University, Newcastle upon Tyne, UK. [4]Recanati Miller Transplantation Institute, Mount Sinai Hospital, New York, NY, USA. [5]These authors contributed equally: Colin Wilson, Samuel Tingle. ✉e-mail: chahanapatel18@gmail.com

According to OPTN data, the non-use rate for recovered pancreases was 23.4% in 2023[3]. Similarly, pancreas utilisation rates are low in the UK, with significant differences in discard rates between centres[4]. Previous efforts to standardise donor selection include the development of the Pancreas Donor Risk Index (PDRI), which aims to grade pancreases based on post-transplant outcomes[5]. However, this focuses on pancreases which have been transplanted rather than guiding pre-transplant decisions. The Pre-Procurement Pancreas Allocation Suitability Score (P-PASS) is designed to assess the suitability of a pancreas for transplantation[6], however, the score was based on expert opinion in 2008, and its applicability has declined as the donor pool has evolved over time. These limitations highlight the need for more evidence-based approaches to improve pancreas utilisation.

A key approach to improving utilisation is through identifying the effects of donor characteristics on pancreas utilisation, allowing for improved donor selection, organ allocation and potential to expand the donor pool. The decision to decline a pancreas for transplant is an inherently multifactorial decision in almost all cases. Even when a primary reason for decline is identified, several other factors likely contributed to the decision-making process. Therefore, the optimal method for assessing the impact of donor factors on utilisation is a modelling approach to utilisation, rather than analysing 'reason for decline' data. Here, we aimed to use a regression modelling approach to assess factors impacting pancreas utilisation in the US, and how these have evolved over time.

In this study, we have found several key variables significantly associated with utilisation, such as age, BMI and donor type. Our analyses reveal increasing reluctance to use DCD and older donors over time, whilst other donor populations, such as Hepatitis C-positive and IVDU donor,s show growing acceptance over time. These findings reveal shifting practices within pancreas transplantation and highlight areas to focus future efforts to improve utilisation both in the US and globally.

## Methods

This was a US-based population cohort study using OPTN data from the United Network for Organ Sharing (UNOS) registry[7]. Specifically, data from the Standard Transplant Analysis and Research files plus the DonorNet supplement were used. All donors in this registry had at least one organ retrieved. Donors declined for all organs, where a retrieval of no organs took place, are not present in the dataset. Given the anonymised nature of the dataset, the NHS Health Research Authority (which oversees Research Ethics Committees, the UK Institutional Review Board equivalent) waived the requirement for Research Ethics Committee approval and informed consent (via the use of their online tool)[8].

We included potential donors who did, or did not, progress to pancreas transplantation between January 2, 2010 and September 30, 2024. We excluded donors with the following characteristics as they are widely deemed absolute contraindications to pancreas transplant in the US: donors without authorisation for pancreas donation, donors with diabetes (or those missing diabetes information), pancreas not used due to surgical trauma, HIV-positive donors and Hepatitis B-positive donors. Pancreases used for islets, and uncontrolled DCD donors were also excluded. Pancreases recovered for research were included in this study as pancreases not used for transplantation, as they are largely offered for clinical transplantation by organ procurement organisation (OPO)s and then declined for transplantation, before being placed for research.

Hepatitis C virus (HCV) positive donors were included in this study, as they are increasingly being considered for transplantation. We defined donors as HCV positive or negative based on their hepatitis C antibody status, as recorded by UNOS. We did not include HCV nucleic antigen tests (NAT) in the definition of HCV status, as this variable was not collected before 2015. We did perform exploratory analyses using the available HCV NAT data.

The primary outcome for this analysis was pancreas utilisation, defined as whether the pancreas was transplanted or not (binary: yes/no). Pancreas non-use included organs that declined for transplantation at any stage of the donation process. This definition does not include offer-level acceptance/decline; a pancreas that is not used will have been declined by all transplant centres to which it was offered to.

## Statistics and Reproducibility

We employed a similar statistical approach to that described previously by our group[9]. To account for missing data, multiple imputation was performed (aregImpute; Hmisc R package in R, version 4.4.2 [R Project for Statistical Computing])to generate 20 imputed datasets[10]. This uses predictive mean matching with bootstrap draws to build rich additive restricted cubic spline models (RCS)[11]. This was preferred over multiple imputation by chained equations, as it preserves non-linear relationships. This was particularly important as our utilisation models applied non-linear modelling. Multiple imputation used all variables listed in Supplementary Data 1, including whether the pancreas was utilised for transplant. We defined rare HLA variants as HLA-A, HLA-B and HLA-DR alleles occurring in less than 1% of the donor population. Hospital stay length was defined as time between date of admission and date of donation.

To assess the association of variables with pancreas utilisation, multivariable logistic regression was performed, pooling results from all 20 imputed datasets, accounting for both within- and between-imputation variance, using the fit.mult.impute function (Hmisc package in R, version 4.4.2)[10]. The event in these models was the transplantation of the pancreas. Restricted cubic splines with 4 knots (5th, 35th, 65th and 95th percentiles) were used to analyse all continuous variables in the main analysis in order to avoid assumptions of linear associations. All statistical tests performed were two-sided.

As a key aim of the study was to assess how risk factors for organ decline have changed over time, an additional model was built incorporating interaction terms between donation date and key variables. Key variables were defined as those with the largest impact on organ non-use in our main multivariable model, determined by the Wald statistic. All continuous variables in these models (including donation date) were modelled with restricted cubic splines with 3 knots (10th, 50th and 90th percentiles); this was selected to prevent overfitting. Interaction terms between two RCS variables (donation date and a continuous key variable) were generally plotted using quartiles of the key variable for illustration. The only exception was donor age, where fixed values of 15, 25, 35 and 45 years were used, as the upper quartiles of donor age were extremely unlikely to be used for transplant.

Donor peak lipase and peak ALT were not included in our main models due to collinearity with donor amylase and AST, respectively. Donor HbA1c was also avoided in the main analysis due to substantial missing data. Sensitivity analyses were performed that included these variables to assess the sensitivity of our models to these decisions.

As the data were clustered by OPO, we built additional models to account for potential OPO effects. Firstly, a hierarchical model was performed with random intercepts for OPO (mixed effects multivariable logistic regression). Secondly, the anonymised OPO code was included as a categorical variable in the main model (fixed effects).

All analyses were performed in R, version 4.4.2 (R Project for Statistical Computing)[12] using the following packages: tidyverse, rms, rmsMD and Hmisc[10,13–15].

## Results

Of the 133,986 potential donors identified between 2010 and 2024, 14,612 (10.9%) pancreases were transplanted. Of the 14,612 pancreas transplants performed, 11,684 were simultaneous pancreas–kidney (SPK), 1723 pancreas transplant alone (PTA) and 1196 pancreas after kidney (PAK) transplants. Information on selection criteria for our cohort is shown in the study flow diagram (Fig. 1). Absolute numbers of pancreas transplants per year decreased steadily over time (Supplementary Fig. 1). In the potential donor cohort, median donor age was 40 years (IQR, 27–53 years), median donor BMI was 26.7 (IQR, 22.9 to 31.3) and the majority of donors were donation after brain death (DBD) (76.5%). 82,039 men (61.2%) and 51,947 women (38.8%) were included in the analysis. Among donors whose

**Fig. 1 | Study fl ow diagram for cohort selection.**
Flowchart outlining inclusion and exclusion criteria
used to defi nestudy cohort.

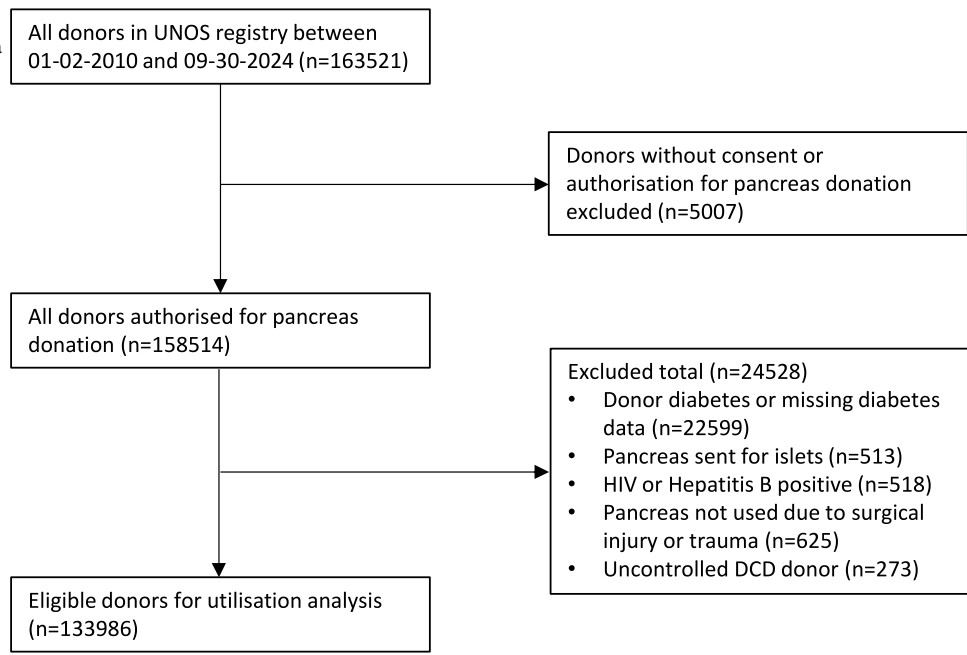

pancreas was utilized for transplant, the median age was 23 years (IQR, 18–29 years), median BMI was 23.4 (IQR, 20.9 to 26.1) and almost all were DBD (97.2%). Key donor demographics are shown in Table 1, with additional demographics and missing data information for all variables given in Supplementary Data 1. As shown in Supplementary Data 1, the proportion of missing data was very low for all donor variables, with peak lipase being the only variable with notable missing data (13.7%).

**Modelling pancreas utilisation decision making**
A multivariable logistic regression model was used to assess the association of various donor characteristics with pancreas utilisation (Table 2, with plots of continuous variables in Fig. 2). These all represent independent associations, adjusted for all other factors in the model. Categorical variables with the strongest (positive or negative) associations with pancreas utilisation were DCD donors, HCV antibody-positive donors and IVDU donors. DCD donors showed 92% lower odds of pancreas utilisation compared to DBD donor utilisation (aOR = 0.078, 95% CI = 0.070 to 0.087, $P = < 0.001$). Hepatitis C positive donors (aOR=0.164, 95% CI = 0.141 to 0.192, $P = < 0.001$) and IVDU donors (aOR=0.572, 95% CI = 0.523 to 0.625, $P = < 0.001$) also showed a significant association with decreased utilisation.

Restricted cubic splines were plotted for all continuous variables in the model shown in Table 2. This non-linear modelling allows the precise nature of these independent associations to be assessed (see Fig. 2). All continuous variables showed significant associations with pancreas utilisation ($P < 0.001$). Donor age and BMI in particular were shown to be significant predictors of pancreas utilisation and demonstrated strong non-linear relationships. Peak pancreas utilisation was seen at a donor age of 27 years, with donors older than 40 years showing a massive reduction utilisation. Donor BMI between 18 and 27 was associated with increased utilisation.

Repeating the model in Table 2 with the inclusion of OPO as a variable (either with a fixed effect per OPO, or in a Hierarchical model with random intercepts for OPO) did not alter these findings. Results from the hier-archical model are shown in Supplementary Table 1 with splines for key variables in Supplementary Fig. 2; the results are entirely in keeping with the main logistic regression model. A sensitivity analysis was performed in which peak lipase and peak ALT were substituted for peak amylase and peak AST (these were not included in the same model due to collinearity). Donor

HbA1c was also adjusted for in this sensitivity analysis due to being excluded from the main analysis for substantial missing data. This showed decreasing pancreas utilisation with peak HbA1c values above 6%. Results from these sensitivity analyses were all consistent with our main analysis and did not alter our conclusions (Supplementary Table 2, splines Supplementary Fig. 3).

Finally, to assess the relative importance of the various donor variables in the model, we calculated Wald statistics. Supplementary Fig. 4 displays the Wald statistics for all included variables, demonstrating the relative importance of each variable as a ranked list. The three most important variables were age, BMI and donor type (DCD versus DBD).

**Trends in utilisation decision-making over time**
We hypothesised that the association of various donor characteristics with pancreas utilisation would have changed over the study duration, as this reflects changes in transplant decision-making over time. We tested this hypothesis by adding interaction terms to the model shown in Table 2. We selected the 10 most important variables from the model shown in Table 2 (ranked list shown in Supplementary Fig. 4), and included interaction terms between these variables and transplant date. The following variables were selected: donor age, BMI, DCD status, HCV antibody status, blood group, history of heavy alcohol use, history of IV drug use, peak creatinine, peak lipase and peak ALT.

The full results from this interaction term model are shown in Supplementary Data 2. Multiple donor characteristics showed significant interactions with donation date. These were assessed by plotting splines of the change in utilisation over time stratified by the key donor variables (as shown in Fig. 3, with additional splines in Supplementary Fig. 5).

Overall, pancreas utilisation has declined over time in the main cohort, reflecting a national downward trend in utilisation. Therefore, when interpreting Fig. 3, it is important to focus on the gap between subgroups to evaluate changing utilisation decision-making, relative to the overall trend of declining utilisation.

The effect of donor age was seen to differ over time (interaction $P < 0.001$). The utilisation of pancreases from younger donors showed minimal change over the study period. However, donors aged 45 were associated with decreasing pancreas utilisation, showing an increased reluctance to accept this population over time, relative to younger donors (Fig. 3A).

**Table 1 | Cohort donor demographics stratified by use of the pancreas for transplantation**

| Donor characteristics | Pancreas not transplanted (N = 119374) | Pancreas transplanted (N = 14612) | Overall (N = 133986) |
|---|---|---|---|
| Age, median [IQR], y | 43.0 [30.0, 55.0] | 23.0 [18.0, 29.0] | 40.0 [27.0, 53.0] |
| BMI, median [IQR], kg/m2 | 27.2 [23.4, 32.0] | 23.4 [20.9, 26.1] | 26.7 [22.9, 31.3] |
| Missing | 315 (0.3%) | 12 (0.1%) | 327 (0.2%) |
| Sex | | | |
| Male | 71935 (60.3%) | 10104 (69.1%) | 82039 (61.2%) |
| Female | 47439 (39.7%) | 4508 (30.9%) | 51947 (38.8%) |
| Ethnicity | | | |
| White | 81880 (68.6%) | 8932 (61.1%) | 90812 (67.8%) |
| Asian | 2778 (2.3%) | 296 (2.0%) | 3074 (2.3%) |
| Black | 16881 (14.1%) | 2868 (19.6%) | 19749 (14.7%) |
| Hispanic/Latino | 16265 (13.6%) | 2344 (16.0%) | 18609 (13.9%) |
| Other | 1491 (1.2%) | 164 (1.1%) | 1655 (1.2%) |
| Missing | 79 (0.1%) | 8 (0.1%) | 87 (0.1%) |
| Cause of death | | | |
| Anoxia | 34286 (28.7%) | 3042 (20.8%) | 37328 (27.9%) |
| Cerebrovascular/ stroke | 33776 (28.3%) | 1550 (10.6%) | 35326 (26.4%) |
| Drug overdose | 16354 (13.7%) | 1516 (10.4%) | 17870 (13.3%) |
| Head trauma | 30935 (25.9%) | 8165 (55.9%) | 39100 (29.2%) |
| Other | 4023 (3.4%) | 339 (2.3%) | 4362 (3.3%) |
| DCD donor | | | |
| No | 88286 (74.0%) | 14210 (97.2%) | 102496 (76.5%) |
| Yes | 31087 (26.0%) | 402 (2.8%) | 31489 (23.5%) |
| Missing | 1 (0.0%) | 0 (0%) | 1 (0.0%) |
| Blood group | | | |
| A | 44486 (37.3%) | 5166 (35.4%) | 49652 (37.1%) |
| AB | 4471 (3.7%) | 219 (1.5%) | 4690 (3.5%) |
| B | 13891 (11.6%) | 1700 (11.6%) | 15591 (11.6%) |
| O | 56517 (47.3%) | 7527 (51.5%) | 64044 (47.8%) |
| Missing | 9 (0.0%) | 0 (0%) | 9 (0.0%) |
| Hepatitis C antibody status | | | |
| Negative | 108632 (91.0%) | 14405 (98.6%) | 123037 (91.8%) |
| Positive | 10715 (9.0%) | 206 (1.4%) | 10921 (8.2%) |
| Missing | 27 (0.0%) | 1 (0.0%) | 28 (0.0%) |
| Hospital stay, median [IQR], days | 4.00 [3.00, 6.00] | 4.00 [3.00, 6.00] | 4.00 [3.00, 6.00] |
| Missing | 583 (0.5%) | 49 (0.3%) | 632 (0.5%) |

Where missing data exists for a variable, this is specified. *BMI* body mass index, *DCD* donation following circulatory death.

There was also significant interaction between donor type and donation date (interaction $P = < 0.001$). Pancreases from DCD donors were consistently associated with lower utilisation than DBD donors, and this gap has continued to increase over the study period (Fig. 3D, F), proving growing reluctance to use DCD pancreases in recent years. DCD pancreas utilisation by centre volume is shown in Supplementary Fig. 6. While some high-volume centres regularly accept DCD donors, the majority of lower-volume centres do not utilise DCD donors at all. The role of normothermic regional perfusion (NRP) on DCD donor utilisation was also assessed. Supplementary Fig. 7 shows counts of DCD donors stratified by retrieval type. Out of 401 DCD donors used for pancreas transplant, 19 were NRP-retrieved (4.74%). The increased use of NRP in the US, has not yet translated to a large number of pancreas transplants from DCD NRP donors (Supplementary Fig. 7).

In contrast, pancreas utilisation from donors with a history of IV drug use increased relative to pancreas utilisation from no-IV drug use donors over the study period. This is evidenced by a tightening of the two curves, which suggests that despite overall declining use of non-IVDU donors, use of IVDU donors has remained stable. This indicates increased acceptance towards the IVDU group, relative to the overall trend.

The impact of HCV antibody status on utilisation changed massively over time (interaction P < 0.001). Prior to 2016, HCV antibody positivity in donors was associated with a drastic decrease in utilisation. Since then, however, growing acceptance of pancreases from HCV antibody-positive donors has increased (Fig. 3C, E), and they are now almost as likely to be utilised as pancreases from HCV antibody-negative donors. This rise is even more pronounced when considered relative to the general decline of 'standard' HCV-negative donors. HCV NAT status was not recorded prior to 2015, and therefore could not be included in our regression models. Supplementary Fig. 8 displays counts and proportions of the HCV antibody-positive donors, stratified by HCV NAT status (missing, positive or negative). This reveals that transplanted HCV antibody-positive donors contain a mix of both NAT-positive and NAT-negative donors (Supplementary Fig. 8).

To better contextualise these observed trends in utilisation, we assessed crude absolute count data for the prevalence of key donor factors over time in both the full donor cohort and in donors used for pancreas transplant (plots in Supplementary Fig. 9 and Supplementary Fig. 10, respectively). We found that the number of older donors and DCD donors increases over time in the full donor population, however this number remains low in the transplanted pancreas cohort. The number of HCV-positive donors and IVDU donors increases over time in both the full and transplanted pancreas cohorts.

## Discussion

This study identified a wide range of significant factors influencing pancreas utilisation, and demonstrated which of these have the largest impact. Our use of non-linear modelling combined with interaction terms allowed us to effectively assess which donor factors impact the decision to use a pancreas, and how this has changed over the study period (2010–2024). This revealed evolving utilisation decision-making processes as the impact of many donor factors on utilisation changed significantly over time.

Prior landmark studies in pancreas transplantation have analysed registry data to investigate predictive factors and post-transplant outcomes[16,17]. In contrast, the methodology used in this study has used registry data to extract insights on the determinants of pancreas utilisation. While this is a less common application of these datasets, it is a demonstration of their capacity to address a diverse range of clinically meaningful questions. The question of utilisation is particularly relevant in the current era, where efforts to increase organ transplantation are a priority. By analysing actual decision-making patterns at a national level, this study reinforces that registry data can yield valuable insights beyond conventional outcome-focused analyses. This analytical approach can be applied beyond pancreas transplantation, allowing for deeper insights into clinical practice that can optimise resource utilisation across specialties. Given the subjective nature of utilisation decisions, qualitative approaches, such as surveys or interviews with clinicians, could provide useful additional perspectives that could add important context to further explain the patterns observed in our analyses.

Previous efforts to guide decision-making within pancreas transplantation have led to the development of several tools, most notably PDRI and P-PASS. The PDRI uses specific donor and transplant factors to predict 1 year graft survival, with a higher PDRI associated with decreased survival. The PDRI has shown variable transferability to non-US populations for predicting post-transplant outcomes[18,19], and has limited use in informing

**Table 2 | Multivariable logistic regression model for pancreas utilisation for the full cohort (n = 133,986)**

| Donor variables | Adjusted Odds Ratio (95% CI) | P-value |
|---|---|---|
| Ethnicity | | |
| White | Ref | |
| Asian, Non-Hispanic | 1.034 (0.889 to 1.202) | 0.668 |
| Black, Non-Hispanic | 1.124 (1.057 to 1.195) | <0.001 |
| Hispanic/Latino | 0.832 (0.783 to 0.885) | <0.001 |
| Other | 0.707 (0.585 to 0.854) | <0.001 |
| Sex: male | 0.994 (0.946 to 1.044) | 0.804 |
| CMV positive | 0.906 (0.866 to 0.948) | <0.001 |
| Blood group | | |
| A | Ref | |
| AB | 0.287 (0.246 to 0.334) | <0.001 |
| B | 0.947 (0.882 to 1.018) | 0.141 |
| O | 1.154 (1.102 to 1.209) | <0.001 |
| Cause of death | | |
| Anoxia | Ref | |
| Cerebrovascular/stroke | 0.987 (0.906 to 1.076) | 0.774 |
| Drug overdose | 1.071 (0.986 to 1.163) | 0.103 |
| Head trauma | 1.068 (1.004 to 1.136) | 0.037 |
| Other | 0.746 (0.647 to 0.860) | <0.001 |
| DCD donor | 0.078 (0.070 to 0.087) | <0.001 |
| Given insulin 24 hours before cross-clamp | 0.941 (0.900 to 0.984) | 0.008 |
| Heavy alcohol use | 0.569 (0.528 to 0.613) | <0.001 |
| Coronary artery disease | 0.389 (0.236 to 0.640) | <0.001 |
| Smoking | 0.847 (0.766 to 0.937) | 0.001 |
| Hypertension | 0.587 (0.532 to 0.647) | <0.001 |
| IV drug use | 0.573 (0.524 to 0.626) | <0.001 |
| Hepatitis C antibody positive | 0.164 (0.140 to 0.191) | <0.001 |
| Number of rare HLA variants | | |
| 0 | Ref | |
| 1 | 0.965 (0.914 to 1.019) | 0.202 |
| 2 | 0.860 (0.721 to 1.025) | 0.143 |
| 3 | 0.744 (0.507 to 1.091) | 0.315 |
| 4 | 0.809 (0.462 to 1.418) | 0.058 |
| 5 | 0.661 (0.453 to 0.965) | 0.032 |
| 6 | 0.793 (0.533 to 1.181) | 0.254 |
| Inotropic support used | 0.968 (0.926 to 1.012) | 0.156 |
| RCS: BMI | RCS terms | <0.001 |
| RCS: Age, years | RCS terms | <0.001 |
| RCS: Donation date | RCS terms | <0.001 |
| RCS: Latest blood pH | RCS terms | <0.001 |
| RCS: Peak lipase | RCS terms | <0.001 |
| RCS: Peak ALT | RCS terms | <0.001 |
| RCS: Peak creatinine | RCS terms | <0.001 |
| RCS: Hospital stay, days | RCS terms | <0.001 |

*P*-values are from two-sided Wald-tests. An event was defined as a pancreas transplant (*n* = 14,612). Results are pooled from 20 imputed datasets. For restricted cubic spline terms, a *p*-value is given for the overall association of the variable with utilisation (Wald-test), plots for these restricted cubic splines are shown in Fig. 2. *CMV* cytomegalovirus, *DCD* donation after circulatory death, *IV* intravenous, *HLA* human leucocyte antigen, *BMI* body mass index, *ALT* alanine aminotransferase.

pre-transplant decision-making. In contrast, P-PASS was designed to guide donor acceptance; however weighting of donor factors was based on expert opinion in 2008, reflecting perceived risk rather than true suitability. As the donor pool has evolved and certain donor characteristics have become more common (such as older donors and DCD donors), the predictive value of P-PASS has decreased[20]. Given these limitations, further approaches are needed to support evidence-based decision-making and reduce pancreas non-use.

We deliberately avoided using 'reason for decline' in our analyses, as this does not capture the multifactorial nature of decision-making. Although a single reason may be recorded, many factors often contribute. For example, a marginal graft may be declined for 'donor body mass index' but might have been accepted had it came from a younger or otherwise optimal donor. By instead using the objective hard endpoint of pancreas use for transplantation as our ground truth, and applying adjusted regression models, we offer a more robust assessment of the factors influencing utilisation that better reflects this complexity.

A key finding of the study is the change in impact of donor type (DBD versus DCD) on pancreas utilisation over time. Several international studies have found similar graft and patient survival rates between DCD and DBD donors[21,22]. DCD pancreases are now being routinely used in Europe, and in some cases, even preferred to DBD donors. In our cohort, 2.8% of transplanted pancreases were from DCD donors, compared to 40% in the UK in 2023/24[23]. Despite this growing body of evidence, we have found that pancreases from DCD donors have been increasingly less likely to be used in comparison to DBD donors in the US. Rather than declining with time, this reluctance to use DCD donors appears to be increasing. The under-utilisation of DCD donors is not distributed evenly as some higher-volume centres show greater acceptance towards DCD donors; however, most lower-volume centres do not use DCD donors at all.

In the full cohort of all potential pancreas donors, there has been a clear rise in the number of DCD donors over time. Despite this greater availability, the number of DCD donors in the transplanted cohort remains consistently low, suggesting many potentially transplantable pancreases may be going unused. Since our data only includes donors from whom at least one organ was recovered, the actual number of potential DCD donors is likely larger than represented here. With DCD donors making up an ever-increasing proportion of the donor pool[24], focusing efforts on broadening acceptance in this cohort could have a significant impact on reducing US transplant waiting lists.

This reluctance to use DCD pancreases appears to have persisted even with the early introduction of NRP, which has been shown to improve both utilisation and outcomes for DCD organs outside of the US[25]. Our results reveal that only a small number of NRP-retrieved pancreases were used for transplant. Widening use of NRP has not yet translated into a significant increase in pancreas utilisation from DCD donors. NRP use in the US was still in its infancy over our study period, potentially limiting its observed impact and highlighting the need for future studies to assess its effect.

Advancements in the medical management of diabetes, such as pump therapy, have likely contributed to greater selectivity in pancreas transplantation, leading to declining transplant numbers. Pancreas transplantation still remains the best treatment option for a select group of patients, as medical management is not universally effective. Demand for pancreas transplants in the US remains high, with just over half of pancreas transplant candidates having waiting times of a year or longer[5]. With earlier transplantation leading to decreased waitlist mortality and greater clinical benefits, evidence-based donor selection is essential. For instance, using a lower-quality DBD donor over a higher-quality DCD donor may lead to poorer outcomes in patients most likely to benefit from transplantation. Therefore, efforts to expand DCD donor utilisation could reduce waitlist pressures and increase timely access to pancreas transplants in the US, where optimising pancreas utilisation has been identified as a national priority[26,27].

A decrease in utilisation of pancreases from older donors was also seen over time. This is likely due to concerns over organ quality due to increased risk of long-term conditions and comorbidities in older individuals.

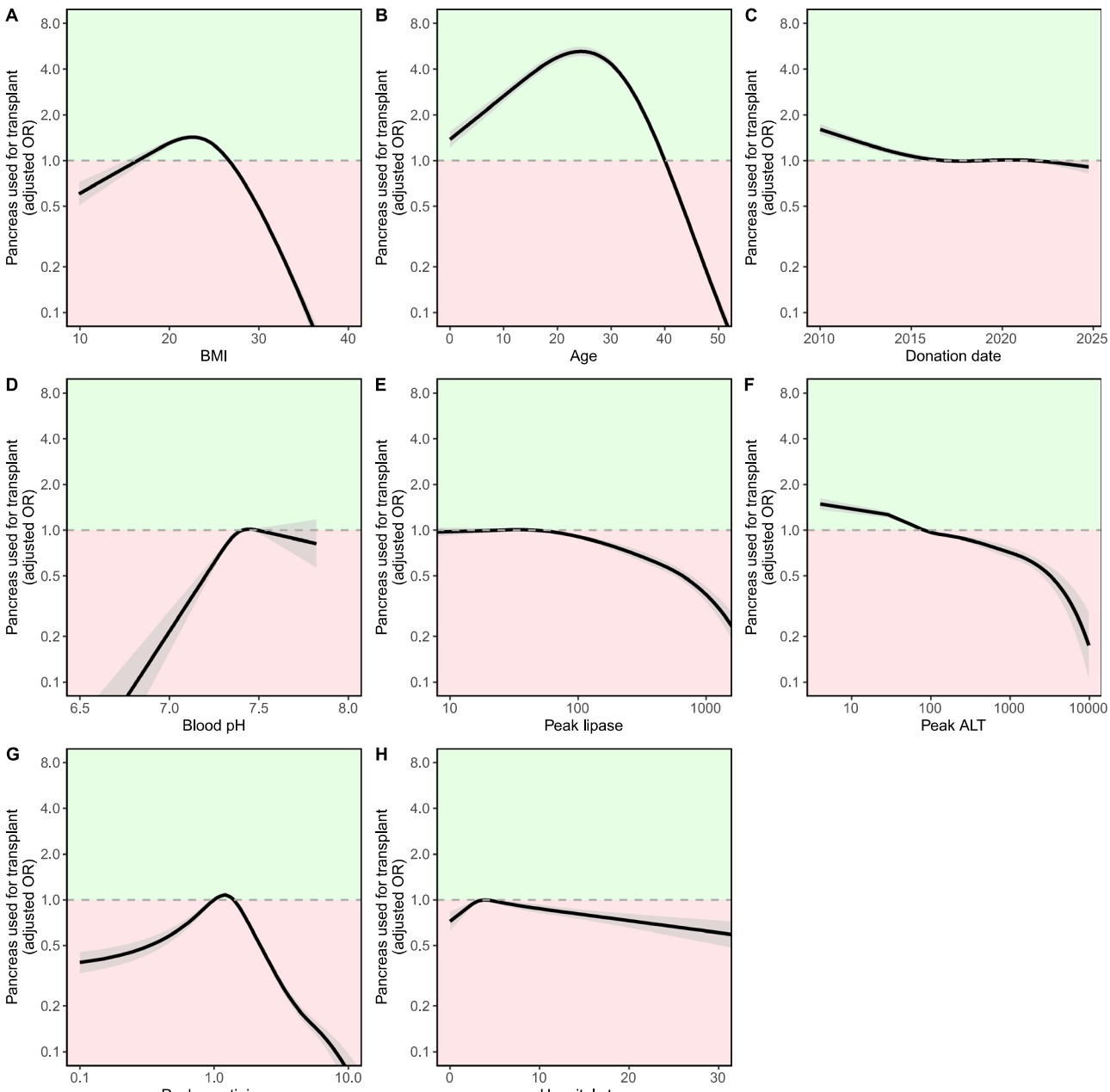

**Fig. 2 | Restricted cubic spline models for non-linear associations between donor factors and pancreas use for transplant.** Plots show association between continuous variables from main logistic regression analysis (Table 2) and pancreas utilisation (*n* = 133,986). Restricted cubic splines were plotted using 4 knots and were adjusted for all variables in Table 2. The solid line represents the estimated odds ratio (OR) for pancreas utilisation relative to the average value of each variable, with grey shading indicating the 95% confidence interval. The dotted horizontal line at OR = 1 represents no effects on utilisation, relative to the reference value. The green shaded area indicates association with increased utilisation, and the red shaded area indicates association with decreased utilisation, relative to the reference value. Donor factors displayed are: **A** BMI. **B** Age. **C** Donation date. **D** Latest blood pH. **E** Peak lipase. **F** Peak ALT. **G** Peak creatinine. **H** Hospital stay length. A log-scaled y-axis is used for all variables for better visualisation of relationships. A log-scaled *x*-axis is also used for peak lipase, peak ALT and peak creatinine. BMI body mass index, ALT alanine aminotransferase.

Emerging evidence suggests that carefully selected pancreases from older donors had comparable outcomes to standard age criteria[28]. Considering the ageing donor population, effectively utilising older donors offers the potential to expand the donor pool.

A donor population where we have demonstrated a significant increase in utilisation over time is HCV antibody-positive donors. HCV antibody-positive donors were consistently associated with non-utilisation before 2016. However, increasing utilisation is seen after 2016, despite a general decline in HCV-negative donor utilisation. HCV antibody-positive donors now see similar pancreas utilisation rates compared to HCV-negative donors, adjusted for other factors. With the widespread use of direct-acting

antiviral therapies, HCV has become a treatable condition, leading to this shift in clinical practice[29]. Although data on HCV NAT were not available for the entire cohort, we have shown that the HCV antibody-positive donors who were the pancreas used for transplant represent a mix of both HCV NAT-positive and negative donors. This reflects utilisation from both donors with active infection and previous or treated infection. The transplant of pancreases from donors with active, NAT-positive, HCV infection represents a major shift in mentality.

In the US, the potential use of HCV-positive donors forms a standard component of the listing and consent process for transplantation. This transparent approach enables wider acceptance and integration of HCV-

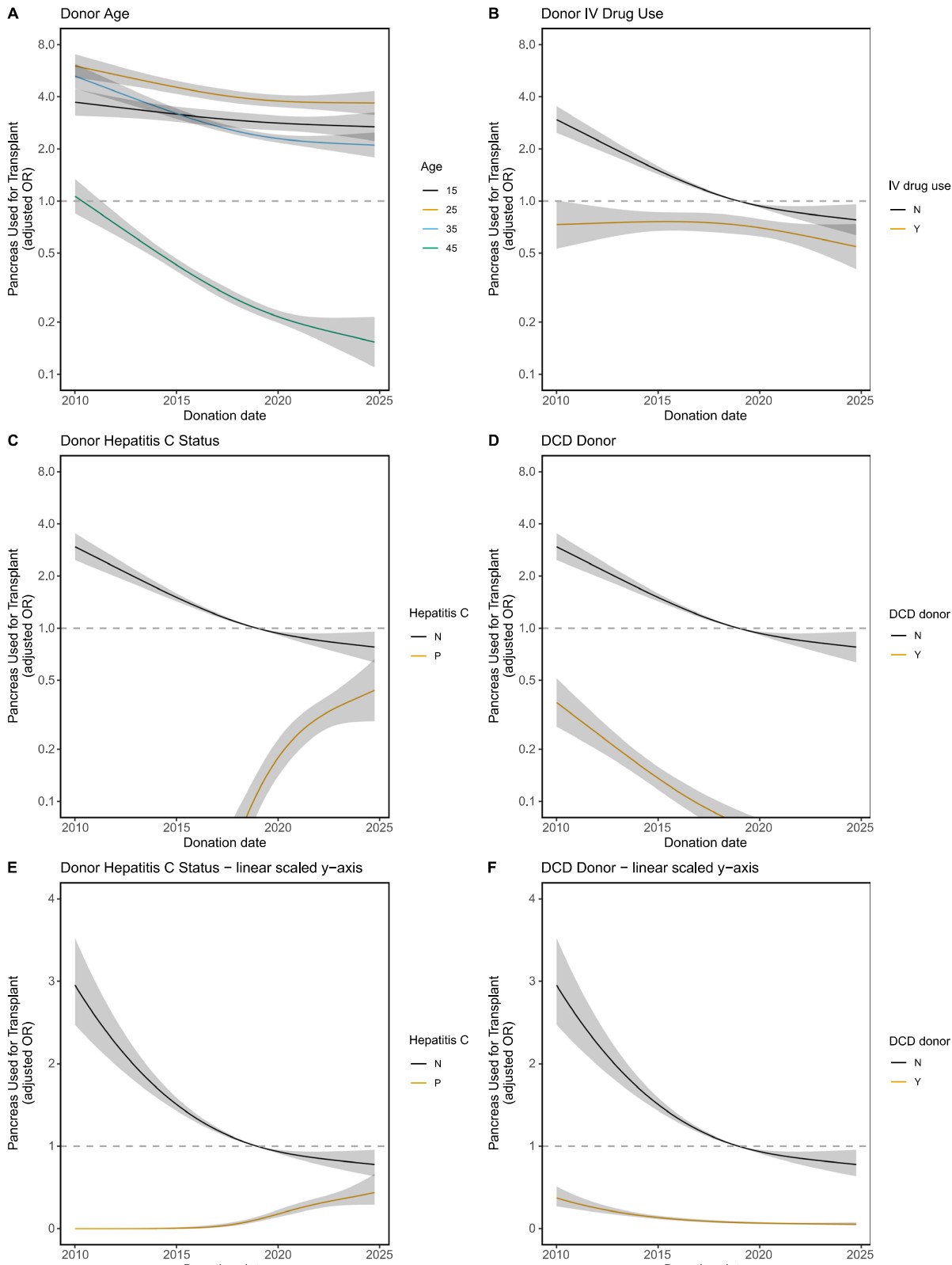

**Fig. 3 | Changes in the impact of key variables on utilisation over time.** Restricted cubic splines (3 knots) illustrating interaction effects between key variables and donation date on pancreas utilisation ($n = 133{,}986$). Grey shading indicates the 95% confidence interval. These are from the model shown in Supplementary Data 2. Additional plots for the remaining key variables are displayed in Supplementary

Fig. 4. The y-axes are on a log-scale, unless otherwise stated. **A** Donor age. **B** Donor IV drug use. **C** Donor hepatitis C antibody status. **D** Donor DCD status. **E** Donor hepatitis C status (linear y-axis). **F** Donor DCD status (linear y-axis). DCD and hepatitis C plots are repeated with a linear y-scale to visualise utilisation when the OR is close to zero. N/Y No/Yes, N/P Negative/Positive.

positive organs into routine practice. In contrast, Europe remains relatively cautious with HCV-positive donors, potentially delaying or limiting the use of high-quality organs. Review of UK data revealed that of the 1818 SPK transplants into adults between 2013 and 2023, only two pancreases were from HCV-positive donors (data provided directly from NHSBT). Our results show that there has been an increase in HCV-positive donors used for pancreas transplant, there has only been a modest rise in HCV-positive donors in the potential donor cohort. The increase in pancreas transplants from hepatitis C-positive donors reflects increasing acceptance of these donors as well as an expansion in the potential donor pool. Implementing a similar approach to the US could normalise the use of this promising new donor source and reduce discards across Europe, improving pancreas utilisation globally.

Another evolving donor population has been IV drug users, with previous biases against this population shifting in recent years. This reflects a wider shift in public health, as deaths from drug overdose have become more common. The US, in particular, has seen a drastic increase in fentanyl-related overdose deaths over this period[30]. Although our analyses adjust for HCV status, increasing use of IVDU donors likely reflects changing perceptions of blood-borne virus risk in IVDU donors, increased ability to treat these viruses, and reduced prejudice against the IVDU population. Whilst these donors are typically younger, our analyses also adjust for donor age, meaning this is not the explanation for the increased use of IVDU donors.

Recent changes to pancreas allocation policy in the US, which were implemented in 2021, have substantially altered pancreas distribution, with a greater number of pancreases now travelling longer distances[31]. These changes aim to improve equitable access to transplantation; however may also impact utilisation due to logistical factors. Since we did not include offer-level data, we could not assess the distance between donor and recipient centres, meaning we cannot comment on the impact of a pancreas being locally recovered or transported on utilisation decisions. Future studies are therefore needed to determine the effect of the new allocation policy on pancreas utilisation, early analyses have shown no statistically significant difference in overall pancreas non-use rate post-implementation[32].

Our analyses include the vast majority of donor characteristics available to transplant centres at the time of organ offer, with the main exception being the visual appearance of the organ. The clinical decision to accept a pancreas is based on these donor factors, which are available at organ offer, and the causal structure of the data is that these donor factors cause differences in utilisation decisions. Recipient factors also contribute to decision-making, this study investigated whether pancreases were accepted for any US candidate, rather than analysing offer-level data.

The main limitation of the study is the retrospective design, we feel this was the optimal study type for our specific research question. We used whether the pancreas was transplanted by any centre as our outcome of interest, and the methodological benefits of this approach are discussed above. This was also chosen as a pancreas transplant by any centre is the key outcome of interest when aiming to increase utilisation. However, as we did not analyse acceptance/decline of individual offers to specific transplant centres/recipients, we cannot comment on individual transplant centre-level practices or potential confounders. We did perform analyses to account for clustering of donors by OPOs and to account for regional variation.

## Conclusions

This study assesses the impact of donor characteristics on pancreas utilisation for transplant. We identified a wide range of significant factors influencing pancreas utilisation, and demonstrated which of these have the largest impact. Combining non-linear modelling and interaction terms allowed us to demonstrate the evolution of utilisation decision-making over time. Notably, we found growing reluctance to use DCD and older donors over time, despite increasing evidence showing favourable outcomes in these groups. This represents a critical area to focus efforts to improve utilisation in the US. In contrast, previously underused groups such as Hepatitis C-positive and IVDU donors show growing acceptance, representing potential new donor sources. These findings should inform and

accelerate efforts to improve pancreas utilisation, both in the US and worldwide.

## Data availability

OPTN data are available upon request to OPTN. These requests may be submitted online, through the following link: https://optn.transplant.hrsa.gov/data/view-data-reports/request-data/. The source data for Fig. 2 and Fig. 3 are provided in Supplementary Data 3 and 4 respectively.

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

## Acknowledgements

SJT was funded for this work via a Medical Research Council Clinical Research Training Fellowship (MRC/Y000676/1), which was part-funded by Kidney Research UK. This study was supported by the National Institute for Health and Care Research (NIHR) Blood and Transplant Research Unit in Organ Donation and Transplantation (NIHR203332), a partnership between NHS Blood and Transplant, University of Cambridge and Newcastle University. The views expressed are those of the author(s) and not necessarily those of the NIHR, NHS Blood and Transplant or the Department of Health and Social Care.

## Author contributions

S.J.T. and C.W. are joint last authors. Study concept: S.J.T. Data acquisition: S.J.T. Data merging and cleaning: C.P., G.K., S.J.T. Statistical analysis: C.P., S.J.T. Interpretation of results: C.P., G.K., Lv.L., M.H., V.W., M.Z.A., S.F., A.M., JS, S.W., C.W., S.J.T. Drafting manuscript: C.P., G.K., LvL, C.W., S.J.T. Critical review of manuscript: C.P., G.K., Lv.L., M.H., V.W., M.Z.A., S.F., A.M., J.S., S.W., C.W., S.J.T.

## Competing interests

The authors declare no competing interests.
