## [Transparent Peer Review file · Communications Medicine]

Modelling donor factors influencing pancreas transplant utilization and evolution of decision-making over time

Corresponding Author: Ms Chahana Patel

Version 0:

Reviewer comments:

Reviewer #1

(Remarks to the Author)

This was a registry analysis of donor factors associated with non-transplantation of pancreas allografts. It is well thought out and well written, a novel approach, and certainly would be of interest to the pancreas transplant community. The methodology appears complex and well executed. I have a few small queries:

1. I was surprised that the entire cohort of potential donors has a median BMI of 26.7 with an upper IQR of 31.3. This seems on the low side in my personal experience. Do the authors think this is because the heavier donors were weeded out with the other exclusions?
2. It would have been nice if the authors had included columns for pancreas transplanted or not in table 1, as they did for the supplementary Table 1
3. I personally disagree with claiming the size of the study based on registry data, as the authors have done in the first sentence of the discussion. That should likely be reserved for when a program is presenting a largest series performed.
4. The authors stop the cohort data just as the allocation in the US has changed. The dynamic of pancreas allocation has certainly changed as well, with many organs now travelling. From the accepting centers' perspective, whether the donor is "local" or "import" may factor into decision making. Is this a factor that the authors may be able to account for throughout the cohort? (or at least comment on in the discussion?)

Reviewer #2

(Remarks to the Author)

Thank you for the opportunity to review this interesting article on the association between donor characteristics and pancreas utilization. In this analysis of pancreas donors using OPTN data from 2010-2024, the authors identify several factors as being especially important for donor utilization, including age, BMI, peak creatinine and donor type. Moreover, the authors demonstrate that the relationship between several of these factors and donor utilization has changed over time, particularly for older age, IV drug use, donor type, and HCV status.

My overall impression is that this article makes a novel and interesting contribution to the literature. However, the authors could make several improvements in service of interpretability, which I detail below.

MAJOR

1. The main source of confusion to me is the description of findings in Figure 3, particularly 3B-F. These constitute some of the main findings of the paper, as the authors highlight in the Results, Discussion, and Abstract that clinicians have become more comfortable transplanting pancreases from HCV+ and IVDU donors. However, the graphs seem to indicate that much of the narrowing in utilization between HCV+ and HCV- donors and between IVDU and non-IVDU donors is due to dramatic decreases in utilization of the "superior" donors (HCV- and non-IVDU). This seems to me to be an important finding, but the authors do not note this pattern or explain why this might be the case. This would imply that rather than an increased willingness to use sub-optimal donors, there has been a decreased willingness to use optimal donors, which would be concerning. It is possible that I am misunderstanding the graphs, in which case the authors can expand and improve upon the description of these figures in the manuscript text.

2. The introduction is currently quite short and would benefit from expansion to better describe the extent of the issue and the gap that this paper fills. For example, what are current discard rates, and how have they changed over time? What does the

existing literature say – including qualitative literature – on decision-making around discard vs. utilization? Are there standard guidelines, and if so what are they, or is there variation by center and by provider? What do we gain from identifying factors associated with donor utilization? All of this would be helpful grounding for the paper and should be accompanied by supporting references.

3. An additional supplementary analysis that would help contextualize the findings would be to simply plot the change in the prevalence of the key factors (age, BMI, HCV status, donor type, IVDU) over time. These could be similar to Supplementary Figure 7. When trying to understand the pattern of increasing HCV+ donor utilization or decreasing DCD donor utilization, it would be helpful to know how much the distribution of HCV+ or DCD among donors has changed over time. For example, does increasing HCV+ donor utilization represent a substantial opportunity, given increasing HCV+ donors? This would be interesting for the authors to dive into in the Discussion.

4. Lines 191-192: “Given the subjective nature of utilization decisions, surveys or interviews with clinicians would fail to fully capture the factors at play” – wouldn’t the subjective nature of decisions make it an ideal area to probe the decision-making thought process through qualitative surveys and interviews with clinicians? I wouldn’t call these qualitative approaches a “limitation” but in fact an integral tool for understanding the associational patterns between the various factors observed in the authors’ findings. Without further probing, the authors observations are just that – observational associations between various characteristics and donor utilization, which may not necessarily reflect the causal factors in clinician decision-making.

MINOR

5. The authors should maintain consistency in abbreviation of US vs. USA, and it should be defined the first time it is used (right now USA appears first in line 43, and then United States in 52, and US in like 62)

6. The Methods section of the Abstract and of the main manuscript should explicitly mention that this study population is US-based

7. Lines 77-84: The authors include a helpful description of imputation of missing data, but they may consider noting here or in the Results when referring to Table 1/Supplementary Table 1 the very low level of missing data for almost all variables (peak lipase being the notable exception)

8. I appreciate the various approaches to modeling clustering by OPO, which is likely an influential level in decisions of donor organ utilization. Are the results of this sensitivity analysis presented anywhere?

9. There is typically no need to present p-values alongside the ORs and 95% CIs, as the p-values do not convey any additional information that is not already conveyed in the 95% CIs. The p-values should therefore be removed from the text of the results, Table 2, and Supplementary Table 2. If the authors wish to present the p-values for the overall association between the variables and utilization for the variables that use restricted cubic splines, I suggest that these be presented in Figure 2. This would also help clarify that the p-values presented in Supplementary Table 3 are for the interaction terms between the variables and donation date (which is in fact different information than is included in the 95% CI).

10. Figure 2 shows very interesting patterns, but the caption could be expanded to improve interpretability. For example, the shaded green and red regions in the graph should be explained.

11. Supplementary Figure 6 – there does appear to be a slight increase in transplanted DCD donors among NRP donors (panel B), despite a very modest increase in number of NRP DCD donors overall (panel A). This stands in contrast to the manuscript text which states that the increased use of NRP has not yet translated to a large number of pancreas transplants from DCD NRP donors (lines 166-167). Would we expect a sharper increase in DCD NRP donor transplants despite the very moderate increase in DCD NRP donors? Put simply, are the authors able to present the % of NRP-retrieved pancreases that were used for transplant?

12. Lines 236-238: Can the authors expand on what the mix of HCV NAT positive and negative donors tells us? The implication is currently unclear

13. Lines 239-243 lack references to support the assertion that Europe remains cautious with HCV+ donors and that discard rates are a problem across Europe

14. Lines 246-248: Given that these are multivariable analyses, the improved treatability of HCV would likely not explain increased use of IVDU donors (which the authors acknowledge with regards to age not explaining the increased use of IVDU donors, given adjustment for age)

15. Lines 250-251: What is it about the retrospective design that the authors feel constitutes the main limitation? What biases are the authors concerned that the retrospective design introduces? One main limitation as I see it is that while the authors conduct multivariable analyses, there is no clear causal structure and we therefore cannot conclude that any given factor has a causal relationship with donor utilization – there may be unmeasured variables or more complex relationships between the variables that explain the observed patterns and drive clinician decision-making.

Version 1:

Reviewer comments:

Reviewer #1

(Remarks to the Author)

Thank you for the thoughtful revisions. I have no further comments.

Reviewer #2

(Remarks to the Author)

The authors have responded to all suggested comments and changes, substantially strengthening the manuscript and improving clarity. I have no further comments, and look forward to seeing this article published.

Point-by-point response

Thank you to the editors and peer reviewers for taking the time to review this manuscript. We have made tracked changes to address these comments, and feel that the manuscript is now much improved as a result. Point-by-point responses to all comments are found below.

Reviewer 1:

This was a registry analysis of donor factors associated with non-transplantation of pancreas allografts. It is well thought out and well written, a novel approach, and certainly would be of interest to the pancreas transplant community. The methodology appears complex and well executed. I have a few small queries:

Thank you for taking the time to review the manuscript, and for these positive comments.

1. I was surprised that the entire cohort of potential donors has a median BMI of 26.7 with an upper IQR of 31.3. This seems on the low side in my personal experience. Do the authors think this is because the heavier donors were weeded out with the other exclusions?

Thank you for highlighting this. Since inclusion in this dataset requires that at least one organ was retrieved by a transplant recovery team, individuals who were declined for all organs before retrieval are not present in the dataset. This likely excludes a proportion of higher-BMI potential donors from the cohort, which may explain why the median BMI appears lower than expected. To clarify this point for readers, we have added the following text to the Methods section (tracked changes line 77-78):

“Donors declined for all organs, where a retrieval of no organs took place, are not present in the dataset.”

2. It would have been nice if the authors had included columns for pancreas transplanted or not in table 1, as they did for the supplementary Table 1

We have now updated Table 1 to include columns for transplanted and non-transplanted pancreases.

3. I personally disagree with claiming the size of the study based on registry data, as the authors have done in the first sentence of the discussion. That should likely be reserved for when a program is presenting a largest series performed.

The Discussion and Conclusion have now been revised to remove claims of this study being “the largest analysis”.

4. The authors stop the cohort data just as the allocation in the US has changed. The dynamic of pancreas allocation has certainly changed as well, with many organs now travelling, From the accepting centers' perspective, whether the donor is "local" or "import" may factor into decision

making. Is this a factor that the authors may be able to account for throughout the cohort? (or at least comment on in the discussion?)

Thank you for highlighting this important point. We agree that the recent pancreas allocation policy changes are likely to influence utilisation and decision-making. We have added text to the discussion acknowledging these allocation changes and highlighting the need for future analyses to assess their impact. We elected to analyse data on whether the organ was accepted by any centre for any recipient, as we feel this is the key metric in addressing utilisation (was the organ used for transplant, or not used for transplant?). This was done in preference to analysing individual offers to individual patients. However, we therefore cannot assess recipient factors (including distance between donor and potential recipient) in these models. Furthermore, individual offer level data (which would be needed to perform an analysis of donor-potential recipient distance) is not available in the UNOS STAR files that were used for this analysis. We now note that variables such as donor and recipient distance, which may influence acceptance decisions under the new allocation system, could not be assessed.

The following paragraph has been added to the Discussion to address this comment (tracked changes line 312-319):

“Recent changes to pancreas allocation policy in the US, which were implemented in 2021, have substantially altered pancreas distribution, with a greater number of pancreases now travelling longer distances.³⁰ These changes aim to improve equitable access to transplantation, however may also impact utilisation due to logistical factors. Since we did not include offer-level data, we could not assess distance between donor and recipient centres, meaning we cannot comment on the impact of a pancreas being locally recovered or transported on utilisation decisions. Future studies are therefore needed to determine the effect of the new allocation policy on pancreas utilisation, however early analyses have shown no statistically significant difference in overall pancreas non-use rate post-implementation.³¹”

30. Organ Procurement and Transplantation Network (OPTN). Removal of DSA and Region from Pancreas Allocation Policy. <https://optn.transplant.hrsa.gov/professionals/by-organ/kidney-pancreas/pancreas-allocation-system/removal-of-dsa-and-region-from-pancreas-allocation-policy/> (2021).

31. Booker, S. E. et al. Impact of removing donation service area and region from pancreas allocation. *Am. J. Transplant.* 24, 1257–1266 (2024).

Reviewer 2:

Thank you for the opportunity to review this interesting article on the association between donor characteristics and pancreas utilization. In this analysis of pancreas donors using OPTN data from 2010-2024, the authors identify several factors as being especially important for donor utilization, including age, BMI, peak creatinine and donor type. Moreover, the authors demonstrate that the relationship between several of these factors and donor utilization has changed over time, particularly for older age, IV drug use, donor type, and HCV status.

My overall impression is that this article makes a novel and interesting contribution to the literature. However, the authors could make several improvements in service of interpretability, which I detail below.

Thank you very much for these encouraging comments.

MAJOR

1. The main source of confusion to me is the description of findings in Figure 3, particularly 3B-F. These constitute some of the main findings of the paper, as the authors highlight in the Results, Discussion, and Abstract that clinicians have become more comfortable transplanting pancreases from HCV+ and IVDU donors. However, the graphs seem to indicate that much of the narrowing in utilization between HCV+ and HCV- donors and between IVDU and non-IVDU donors is due to dramatic decreases in utilization of the “superior” donors (HCV- and non-IVDU). This seems to me to be an important finding, but the authors do not note this pattern or explain why this might be the case. This would imply that rather than an increased willingness to use sub-optimal donors, there has been a decreased willingness to use optimal donors, which would be concerning. It is possible that I am misunderstanding the graphs, in which case the authors can expand and improve upon the description of these figures in the manuscript text.

Thank you for this comment. We agree that the trends in Figure 3 require clearer explanation in the manuscript. There has been a general trends towards a decrease in utilisation over time, which means changes in utilisation of donor factors must be interpreted relative to this overall decline. For IVDU donors, utilisation remained stable, whereas utilisation among non-IVDU donors has declined, resulting in a narrowing difference, and indicating increased willingness to use IVDU donors (relative to ‘standard’ non-IVDU donors). For HCV-positive donors, utilisation has increased over time. When compared with the decline in HCV-negative donor utilisation, this represents a substantial relative increase in acceptance of HCV-positive pancreases. An relative increase which is contributed to both by an absolute increase in the number of HCV positive donors where the pancreas is used, as well as falling utilisation in the overall cohort.

We have now revised the Results text to emphasise the importance of interpreting these trends relative to the background decline in utilisation. The following text has been added to better explain the observed results (tracked changes line 182-184, line 199-201 and line 205-206):

- *“Overall, pancreas utilisation has declined over time in the main cohort, reflecting a national downward trend in utilisation. Therefore, when interpreting Figure 3, it is important to focus on the gap between subgroups to evaluate changing utilisation decision making, relative to the overall trend of declining utilisation.”*
- *“This is evidenced by a tightening of the two curves, which suggests that despite overall declining use of non-IVDU donors, use of IVDU donors has remained stable. This indicates increased acceptance towards the IVDU group, relative to the overall trend.”*
- *“This rise is even more pronounced when considered relative to general decline of ‘standard’ HCV-negative donors.”*

2. The introduction is currently quite short and would benefit from expansion to better describe the extent of the issue and the gap that this paper fills. For example, what are current discard rates, and how have they changed over time? What does the existing literature say – including qualitative literature – on decision-making around discard vs. utilization? Are there standard guidelines, and if so what are they, or is there variation by center and by provider? What do we

gain from identifying factors associated with donor utilization? All of this would be helpful grounding for the paper and should be accompanied by supporting references.

Thank you for this comment. We have expanded the introduction to better describe non-use rates, existing tools to guide decision-making and their limitations.

The following paragraph has been added to the introduction to address this comment with corresponding references (tracked changes line 54-61):

“According to OPTN data, the non-use rate for recovered pancreases was 23.4% in 2023.³ Similarly, pancreas utilisation rates are low in the UK with significant differences in discard rates between centres.⁴ Previous efforts to standardise donor selection include the development of the Pancreas Donor Risk Index (PDRI) which aims to grade pancreases based on post-transplant outcomes.⁵ However, this focuses on pancreases which have been transplanted rather than guiding pre-transplant decisions. The Pre-Procurement Pancreas Allocation Suitability Score (P-PASS) is designed to assess suitability of a pancreas for transplantation,⁶ however the score was based on expert opinion in 2008, and its applicability has declined as the donor pool has evolved over time. These limitations highlight the need for more evidence-based approaches to improve pancreas utilisation.”

3. Raja Kandaswamy et al. OPTN/SRTR 2023 annual data report: pancreas. *Am. J. Transplant.* 25, S138–192 (2025).
4. Cornateanu, S. M. et al. Pancreas utilization rates in the UK – an 11-year analysis. *Transpl. Int.* 34, 1306–1318 (2021).
5. Axelrod, D.A., Sung, R.S., Meyer, K.H., Wolfe, R.A. & Kaufman, D.B. Systematic evaluation of pancreas allograft quality, outcomes and geographic variation in utilization. *Am. J. Transplant.* 10, 837–845 (2010).
6. Vinkers, M. T., Rahmel, A. O., Slot, M. C., Smits, J. M. & Schareck, W. D. How to recognize a suitable pancreas donor: a Eurotransplant study of pre-procurement factors. *Transplant. Proc.* 40, 1275–1278 (2008)

3. An additional supplementary analysis that would help contextualize the findings would be to simply plot the change in the prevalence of the key factors (age, BMI, HCV status, donor type, IVDU) over time. These could be similar to Supplementary Figure 7. When trying to understand the pattern of increasing HCV+ donor utilization or decreasing DCD donor utilization, it would be helpful to know how much the distribution of HCV+ or DCD among donors has changed over time. For example, does increasing HCV+ donor utilization represent a substantial opportunity, given increasing HCV+ donors? This would be interesting for the authors to dive into in the Discussion.

Thank you for this comment. We have now added Supplementary Figure 9 and Supplementary Figure 10 to our Results section which display prevalence of key donor factors over time in both the full donor cohort and pancreas transplanted cohort respectively. The following paragraph has been added to the Results section (tracked changes line 211-216):

“To better contextualise these observed trends in utilisation, we assessed crude absolute count data for the prevalence of key donor factors over time in both the full donor cohort and in donors used for pancreas transplant (plots in Supplementary Figure 9 and Supplementary Figure 10 respectively). We found that the number of older donors and DCD donors increases over time in the full donor population, however this number remains low in the transplanted pancreas cohort. The number of

HCV-positive donors and IVDU donors increases over time in both the full and transplanted pancreas cohorts.”

The following points have been added to the Discussion to better contextualise our findings given the results of these supplementary analyses (tracked changes line 258-262 and line 300-303):

- *“In the full cohort of all potential pancreas donors, there has been a clear rise in the number of DCD donors over time. Despite this greater availability, the number of DCD donors in the transplanted cohort remains consistently low, suggesting many potentially transplantable pancreases may be going unused. Since our data only includes donors from whom at least one organ was recovered, the actual number of potential DCD donors is likely larger than represented here.”*
- *“Our results show that there has been an increase in HCV-positive donors used for pancreas transplant, however there has only been a modest rise in HCV-positive donors in the potential donor cohort. The increase in pancreas transplants from hepatitis C positive donors reflects increasing acceptance of these donors as well as an expansion in the potential donor pool.”*

4. Lines 191-192: “Given the subjective nature of utilization decisions, surveys or interviews with clinicians would fail to fully capture the factors at play” – wouldn’t the subjective nature of decisions make it an ideal area to probe the decision-making thought process through qualitative surveys and interviews with clinicians? I wouldn’t call these qualitative approaches a “limitation” but in fact an integral tool for understanding the associational patterns between the various factors observed in the authors’ findings. Without further probing, the authors observations are just that – observational associations between various characteristics and donor utilization, which may not necessarily reflect the causal factors in clinician decision-making.

Thank you for highlighting this. We agree that qualitative approaches could provide additional insights into clinical decision making and have added the following sentence to the Discussion (tracked changes line 231-233):

“Given the subjective nature of utilisation decisions, qualitative approaches, such as surveys or interviews with clinicians, could provide useful additional perspectives which could add important context to further explain the patterns observed in our analyses.”

We have also added clarification regarding the proposed causal structure of the data to the Discussion (tracked changes line 320-325):

“Our analyses include the vast majority of donor characteristics available to transplant centres at the time of organ offer, with the main exception being visual appearance of the organ. The clinical decision to accept a pancreas is based on these donor factors which are available at organ offer, and the causal structure of the data is that these donor factors cause differences in utilisation decisions. Recipient factors also contribute to decision-making, however this study investigated whether pancreases were accepted for any US candidate, rather than analysing offer-level data.”

MINOR

5. The authors should maintain consistency in abbreviation of US vs. USA, and it should be defined

the first time it is used (right now USA appears first in line 43, and then United States in 52, and US in like 62)

Thank you for highlighting this inconsistency. We have updated the manuscript to define the abbreviation (tracked change line 22) and consistently use “US”.

6. The Methods section of the Abstract and of the main manuscript should explicitly mention that this study population is US-based

The Methods sections have now been updated to explicitly mention this is a US-based study (tracked changes line 22 and 75).

7. Lines 77-84: The authors include a helpful description of imputation of missing data, but they may consider noting here or in the Results when referring to Table 1/Supplementary Table 1 the very low level of missing data for almost all variables (peak lipase being the notable exception)

Thank you for highlighting this. We have added the following text to the Results when referring to these tables (tracked changes line 139-141):

“As shown in Supplementary Table 1, the proportion of missing data was very low for all donor variables, with peak lipase being the only variable with notable missing data (13.7%).”

8. I appreciate the various approaches to modeling clustering by OPO, which is likely an influential level in decisions of donor organ utilization. Are the results of this sensitivity analysis presented anywhere?

Thank you for this comment. We have now included the results of this sensitivity analysis in Supplementary Table 2 and Supplementary Figure 2. Supplementary Table 2 confirms that the fixed effects are virtually identical in a logistic regression model, and a mixed effects model with a random intercept for OPO. We have also added the following text to the Results (tracked changes line 158-160):

“Results from the hierarchical model are shown in Supplementary Table 2 with splines for key variables in Supplementary Figure 2; the results are entirely in keeping with the main logistic regression model.”

9. There is typically no need to present p-values alongside the ORs and 95% CIs, as the p-values do not convey any additional information that is not already conveyed in the 95% CIs. The p-values should therefore be removed from the text of the results, Table 2, and Supplementary Table 2. If the authors wish to present the p-values for the overall association between the variables and utilization for the variables that use restricted cubic splines, I suggest that these be presented in Figure 2. This would also help clarify that the p-values presented in Supplementary Table 3 are for the interaction terms between the variables and donation date (which is in fact different information than is included in the 95% CI).

Thank you for highlighting this. We felt that the p-values are useful to readers in scanning the table. We therefore reached out to the editor after reading your comment, and the editorial team has advised that we keep the p-values in their current format. Regarding clarification of p-values in

Supplementary Table 3, we have updated “RCS terms” to “RCS interaction terms” to clarify these represent interaction terms, as suggested in your comment.

10. Figure 2 shows very interesting patterns, but the caption could be expanded to improve interpretability. For example, the shaded green and red regions in the graph should be explained.

Thank you for highlighting this. We have expanded the caption for Figure 2, and other figures, to include explanation of the shaded 95% CI, red and green shading and the dotted horizontal line. We have also clarified the solid line represents OR relative to the average value of each variable. The caption for Figure 2 is pasted below:

“Figure 2: Restricted cubic spline models showing association between continuous variables from main logistic regression analysis (Table 2) and pancreas utilisation. Restricted cubic splines were plotted using 4 knots and were adjusted for all variables in Table 2. The solid line represents the estimated odds ratio (OR) for pancreas utilisation relative to the average value of each variable, with grey shading indicating the 95% confidence interval. The dotted horizontal line at OR=1 represents no effects on utilisation, relative to the reference value. The green shaded area indicates association with increased utilisation, and the red shaded area indicates association with decreased utilisation, relative to the reference value. Donor factors displayed are: A) BMI. B) Age. C) Donation date. D) Latest blood pH. E) Peak lipase. F) Peak ALT. G) Peak creatinine. H) Hospital stay length. A log-scaled y-axis is used for all variables for better visualisation of relationships. A log-scaled x-axis is also used for peak lipase, peak ALT and peak creatinine. Abbreviations: BMI = body mass index, ALT = alanine aminotransferase.”

11. Supplementary Figure 6 – there does appear to be a slight increase in transplanted DCD donors among NRP donors (panel B), despite a very modest increase in number of NRP DCD donors overall (panel A). This stands in contrast to the manuscript text which states that the increased use of NRP has not yet translated to a large number of pancreas transplants from DCD NRP donors (lines 166-167). Would we expect a sharper increase in DCD NRP donor transplants despite the very moderate increase in DCD NRP donors? Put simply, are the authors able to present the % of NRP-retrieved pancreases that were used for transplant?

Thank you for this comment. Our intention was to convey that, although there has been a rise in DCD NRP donors, increased NRP use has not yet resulted in a substantial increase in pancreas utilisation from DCD donors. Since NRP is still in its relative infancy in the US, it is important to assess its impact with future studies.

To address this comment, the following sentence has been added to the Discussion (tracked changes line 267-269):

“NRP use in the US was still in its infancy over our study period, potentially limiting its observed impact and highlighting the need for future studies to assess its effect.”

We have also added the percentage of NRP-retrieved pancreases that were transplanted to the Results (tracked changes line 195-196):

“Out of 401 DCD donors used for pancreas transplant, 19 were NRP-retrieved (4.74%).”

12. Lines 236-238: Can the authors expand on what the mix of HCV NAT positive and negative donors tells us? The implication is currently unclear

Thank you for highlighting this. We included HCV NAT data as a supplementary analysis since this variable was not available for the first 5 years of study period. We have now added text (tracked changes line 292-294) to explain that the observed mix of NAT-positive and NAT-negative donors indicates that pancreases are being used from donors with both active HCV infection and previous/treated infection. This distinction is important, as active infection may be perceived as carrying higher risk for viral transmission to the recipient by clinicians. The added text is pasted below:

“This reflects utilisation from both donors with active infection and previous or treated infection. The transplant of pancreases from donors with active, NAT positive, HCV infection represents a major shift in mentality.”

13. Lines 239-243 lack references to support the assertion that Europe remains cautious with HCV+ donors and that discard rates are a problem across Europe

Thank you for highlighting this important point. We have struggled to find a suitable reference on donor HCV positive pancreas transplants across Europe, as it is so infrequently performed that there are no reports available. To address your comment and provide evidence for this statement, we have assessed the UK Transplant Registry data and the following has been added to the Discussion (tracked changes line 298-299):

“Review of UK data revealed that of the 1818 SPK transplants into adults between 2013 and 2023, only two pancreases were from HCV-positive donors (data provided directly from NHSBT).”

14. Lines 246-248: Given that these are multivariable analyses, the improved treatability of HCV would likely not explain increased use of IVDU donors (which the authors acknowledge with regards to age not explaining the increased use of IVDU donors, given adjustment for age)

Thank you for this comment. Adjustment for HCV only adjusts for those donors where HCV infection was confirmed. Many IVDU who are HCV negative may still be considered higher risk due to possible window period infections. Therefore, the improved treatability of HCV may make these donors more acceptable in general (including those without HCV positive antibody/NAT), an effect that would not be captured by adjustment. We have added the following for clarification (tracked changes line 307-309):

“Although our analyses adjust for HCV status, increasing use of IVDU donors likely reflects changing perceptions of blood-borne virus risk in IVDU donors, increased ability to treat these viruses, and reduced prejudice against the IVDU population.”

15. Lines 250-251: What is it about the retrospective design that the authors feel constitutes the main limitation? What biases are the authors concerned that the retrospective design introduces? One main limitation as I see it is that while the authors conduct multivariable analyses, there is no clear causal structure and we therefore cannot conclude that any given factor has a causal relationship with donor utilization – there may be unmeasured variables or more complex

relationships between the variables that explain the observed patterns and drive clinician decision-making.

Thank you for highlighting this important point. We agree that there may be unmeasured variables or more complex relationships between variables that explain the observed patterns and drive decision-making. However, our models include the vast majority of donor factors available to transplant centres at the time of organ offer, with the main exception being visual appearance. Since transplant decisions are based on these factors, the causal structure of the data is that these donor characteristics cause differences in utilisation decisions. Since our study did not include offer level data, we cannot comment on recipient factors that also contribute to decision-making.

The following paragraph has been added to the Discussion to address this comment (tracked changes line 320-325):

“Our analyses include the vast majority of donor characteristics available to transplant centres at the time of organ offer, with the main exception being visual appearance of the organ. The clinical decision to accept a pancreas is based on these donor factors which are available at organ offer, and the causal structure of the data is that these donor factors cause differences in utilisation decisions. Recipient factors also contribute to decision-making, however this study investigated whether pancreases were accepted for any US candidate, rather than analysing offer-level data.”